# XCiT: Cross-Covariance Image Transformers

Alaaeldin El-Nouby[1,2]    Hugo Touvron[1,3]    Mathilde Caron[1,2]    Piotr Bojanowski[1]

Matthijs Douze[1]    Armand Joulin[1]    Ivan Laptev[2]    Natalia Neverova[1]

Gabriel Synnaeve[1]    Jakob Verbeek[1]    Hervé Jégou[1]

[1]Facebook AI    [2]Inria    [3]Sorbonne University

## Abstract

Following tremendous success in natural language processing, transformers have recently shown much promise for computer vision. The self-attention operation underlying transformers yields global interactions between all tokens, *i.e.* words or image patches, and enables flexible modelling of image data beyond the local interactions of convolutions. This flexibility, however, comes with a quadratic complexity in time and memory, hindering application to long sequences and high-resolution images. We propose a "transposed" version of self-attention that operates across feature channels rather than tokens, where the interactions are based on the cross-covariance matrix between keys and queries. The resulting cross-covariance attention (XCA) has linear complexity in the number of tokens, and allows efficient processing of high-resolution images. Our cross-covariance image transformer (XCiT) – built upon XCA – combines the accuracy of conventional transformers with the scalability of convolutional architectures. We validate the effectiveness and generality of XCiT by reporting excellent results on multiple vision benchmarks, including (self-supervised) image classification on ImageNet-1k, object detection and instance segmentation on COCO, and semantic segmentation on ADE20k.

## 1 Introduction

Transformers architectures [68] have provided quantitative and qualitative breakthroughs in speech and natural language processing (NLP). After a few attempts to incorporate wide-range self-attention in vision architectures [71, 82], Dosovitskiy et al. [21] established transformers as a viable architecture for learning visual representations, reporting competitive results for image classification while relying on large-scale pre-training. Touvron et al. [64] have shown on par or better accuracy/throughput compared to strong convolutional baselines such as EfficientNets [58] when training transformers on ImageNet-1k using extensive data augmentation and improved training schemes. Promising results have been obtained for other vision tasks, including image retrieval [22], object detection and semantic segmentation [44, 70, 81, 83], as well as video understanding [2, 7, 23].

One major drawback of transformers is the time and memory complexity of the core self-attention operation, that increases quadratically with the number of input tokens, or similarly number of patches in computer vision. For $w \times h$ images, this translates to a complexity of $\mathcal{O}(w^2 h^2)$, which is prohibitive for most tasks involving high-resolution images, such as object detection and segmentation. Various strategies have been proposed to alleviate this complexity, for instance using approximate forms of self-attention [44, 81], or pyramidal architectures which progressively downsample the feature maps [70]. However, none of the existing solutions are fully satisfactory, as they either trade complexity for accuracy, or their complexity remains excessive for processing very large images.

We replace the self-attention, as originally introduced by Vaswani et al. [68], with a "transposed" attention that we denote as "cross-covariance attention" (XCA). Cross-covariance attention substi-

---

Code: `https://github.com/facebookresearch/xcit`

35th Conference on Neural Information Processing Systems (NeurIPS 2021).

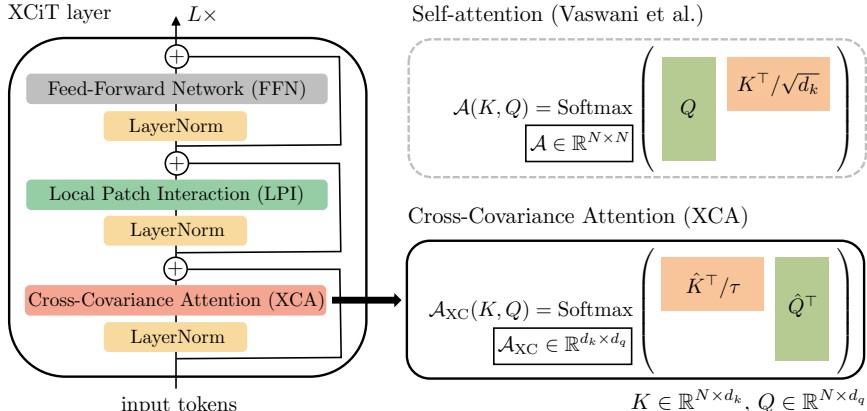

Figure 1: Our XCiT layer consists of three main blocks, each preceded by LayerNorm and followed by a residual connection: (i) the core cross-covariance attention (XCA) operation, (ii) the local patch interaction (LPI) module, and (iii) a feed-forward network (FFN). By transposing the query-key interaction, the computational complexity of XCA is linear in the number of data elements $N$, rather than quadratic as in conventional self-attention.

tutes the explicit full pairwise interaction between tokens by self-attention among features, where the attention map is derived from the cross-covariance matrix computed over the key and query projections of the token features. Importantly, XCA has a linear complexity in the number of patches. To construct our Cross-Covariance Image Transformers (XCiT), we combine XCA with local patch interaction modules that rely on efficient depth-wise convolutions and point-wise feedforward networks commonly used in transformers, see Figure 1. XCA can be regarded as a form of a dynamic $1 \times 1$ convolution, which multiplies all tokens with the same data-dependent weight matrix. We find that the performance of our XCA layer can be further improved by applying it on blocks of channels, rather than directly mixing all channels together. This "block-diagonal" shape of XCA further reduces the computational complexity with a factor linear in the number of blocks.

Given its linear complexity in the number of tokens, XCiT can efficiently process images with more than thousand pixels in each dimension. Notably, our experiments show that XCiT does not compromise the accuracy and achieves similar results to DeiT [64] and CaiT [67] in comparable settings. Moreover, for dense prediction tasks such as object detection and image segmentation, our models outperform popular ResNet [28] backbones as well as the recent transformer-based models [44, 70, 81]. Finally, we also successfully apply XCiT to the self-supervised feature learning using DINO [12], and demonstrate improved performance compared to a DeiT-based backbone [64].

Overall, we summarize our contributions as follows:

- We introduce cross-covariance attention (XCA), which provides a "transposed" alternative to conventional self-attention, attending over channels instead of tokens. Its complexity is linear in the number of tokens, allowing for efficient processing of high-resolution images, see Figure 2.

- XCA attends to a fixed number of channels, irrespective of the number of tokens. As a result, our models are significantly more robust to changes in image resolution at test time, and are therefore more amenable to process variable-size images.

- For image classification, we demonstrate that our models are on par with state-of-the-art vision transformers for multiple model sizes using a simple columnar architecture, *i.e.*, in which we keep the resolution constant across layers. In particular, our XCiT-L24 model achieves 86.0% top-1 accuracy on ImageNet, outperforming its CaiT-M24 [67] and NFNet-F2 [10] counterparts with comparable numbers of parameters.

- For dense prediction tasks with high-resolution images, our models outperform ResNet and multiple transformer-based backbones. On the COCO benchmark, we achieve a strong performance of 48.5% and 43.7% mAP for object detection and instance segmentation respectively. Moreover, we report 48.4% mIoU for semantic segmentation on the ADE20k benchmark, outperforming the state-of-the-art Swin Transformer [44] backbones across all comparable model sizes.

- Finally, our XCiT model is highly effective in self-supervised learning setups, achieving 80.9% top-1 accuracy on ImageNet-1k using DINO [12].

## 2 Related work

**Deep vision transformers.** Training deep vision transformers can be challenging due to instabilities and optimization issues. Touvron et al. [67] successfully train models with up to 48 layers using LayerScale, which weighs contributions of residual blocks across layers and improves optimization. Additionally, the authors introduce class attention layers which decouple the learning of patch features and the feature aggregation stage for classification.

**Spatial structure in vision transformers.** Yuan et al. [79] propose applying a soft split for patch projection with overlapping patches which is applied repeatedly across model layers, reducing the number of patches progressively. Han et al. [27] introduce a transformer module for intra-patch structure, exploiting pixel-level information and integrating with an inter-patch transformer to attain higher representation power. d'Ascoli et al. [18] consider the initialization of self-attention blocks as a convolutional operator, and demonstrate that such initialization improves the performance of vision transformers in low-data regimes. Graham et al. [26] introduce LeViT, which adopts a multi-stage architecture with progressively reduced feature resolution similar to popular convolutional architectures, allowing for models with high inference speed while retaining a strong performance. Moreover, the authors adopt a convolution-based module for extracting patch descriptors. Yuan et al. [78] improve both the performance and the convergence speed of vision transformers by replacing the linear patch projection with convolutional layers and max-pooling, as well as modifying the feed-forward networks in each transformer layer to incorporate depth-wise convolutions.

**Efficient attention.** Numerous methods for efficient self-attention have been proposed in the literature to address the quadratic complexity of self-attention in the number of input tokens. These include restricting the span of the self-attention to local windows [48, 50], strided patterns [14], axial patterns [30], or an adaptive computation across layers [57]. Other methods provide an approximation of the self-attention matrix which can be achieved by a projection across the token dimension [69], or through a factorization of the softmax-attention kernel [15, 37, 56, 77], which avoids explicit computation of the attention matrix. While conceptually different, our XCA performs similar computations without being sensitive to the choice of the kernel. Similarly, Lee-Thorp et al. [41] achieve faster training by substituting self-attention with unparametrized Fourier Transform. Other efficient attention methods rely on local attention and adding a small number of global tokens, thus allowing interaction among all tokens only by hopping through the global tokens [1, 5, 34, 80]. Similarly, Goyal et al. [25] use a global workspace though which items interact, albeit one that is shared across layers.

**Transformers for high-resolution images.** Several works adopt visual transformers to high-resolution image tasks beyond image classification, such as object detection and image segmentation. Wang et al. [70] design a model with a pyramidal architecture and address complexity by gradually reducing the spatial resolution of keys and values. Similarly, for video recognition Fan et al. [23] utilize pooling to reduce the resolution across the spatial and temporal dimensions to allow for an efficient computation of the attention matrix. Zhang et al. [81] adopt global tokens and local attention to reduce the model complexity, while Liu et al. [44] provide an efficient method for local attention with shifted windows. In addition, Zheng et al. [83] and Ranftl et al. [54] study problems like semantic segmentation and monocular depth estimation with the quadratic self-attention operation.

**Data-dependent layers.** Our XCiT layer can be regarded as a "dynamic" $1\times1$ convolution, which multiplies all token features with the same data-dependent weight matrix, derived from the key and query cross-covariance matrix. In the context of convolutional networks, Dynamic Filter Networks [9] explore a related idea, using a filter generating subnetwork to produce convolutional filters based on features in previous layers. Squeeze-and-Excitation networks [32] use data dependent $1 \times 1$ convolutions in convolutional architectures. Spatially average-pooled features are fed to a 2-layer MLP which produces per channel scaling parameters. Closer in spirit to our work, Lambda layers propose a way to ensure global interaction in ResNet models [4]. Their "content-based lambda function" is computing a similar term as our cross-covariance attention, but differing in how the softmax and $\ell_2$ normalizations are applied. Moreover, Lambda layers also include specific position-based lambda functions, and LambdaNetworks are based on ResNets while XCiT follows the ViT architecture. Recently *data-independent* analogues of self-attention have also been found to be an effective alternative to convolutional and self-attention layers for vision tasks [20, 46, 62, 66]. These

methods treat entries in the attention map as learnable parameters, rather than deriving the attention map dynamically from queries and keys, but their complexity remains quadratic in the number of tokens. Zhao *et al.* [82] consider alternative attention forms in computer vision.

# 3   Method

In this section, we first recall the self-attention mechanism, and the connection between the Gram and covariance matrices, which motivated our work. We then propose our cross-covariance attention operation (XCA) – which operates along the feature dimension instead of token dimension in conventional transformers – and combine it with local patch interaction and feedforward layers to construct our Cross-Covariance Image Transformer (XCiT). See Figure 1 for an overview.

## 3.1   Background

**Token self-attention.**   Self-attention, as introduced by Vaswani et al. [68], operates on an input matrix $X \in \mathbb{R}^{N \times d}$, where $N$ is the number of tokens, each of dimensionality $d$. The input $X$ is linearly projected to queries, keys and values, using the weight matrices $W_q \in \mathbb{R}^{d \times d_q}$, $W_k \in \mathbb{R}^{d \times d_k}$ and $W_v \in \mathbb{R}^{d \times d_v}$, such that $Q{=}XW_q$, $K{=}XW_k$ and $V{=}XW_v$, where $d_q = d_k$. Keys and values are used to compute an attention map $\mathcal{A}(K, Q) = \mathrm{Softmax}(QK^\top / \sqrt{d_k})$, and the output of the self-attention operation is defined as the weighted sum of $N$ token features in $V$ with the weights corresponding to the attention map: $\mathrm{Attention}(Q, K, V) = \mathcal{A}(K, Q)V$. The computational complexity of self-attention scales quadratically in $N$, due to pairwise interactions between all $N$ elements.

**Relationship between Gram and covariance matrices.**   To motivate our cross-covariance attention operation, we recall the relation between Gram and covariance matrices. The unnormalised $d \times d$ covariance matrix is obtained as $C{=}X^\top X$. The $N \times N$ Gram matrix contains all pairwise innerproducts: $G{=}XX^\top$. The non-zero part of the eigenspectrum of the Gram and covariance matrix are equivalent, and the eigenvectors of $C$ and $G$ can be computed in terms of each other. If $V$ are the eigenvectors of $G$, then the eigenvectors of $C$ are given by $U{=}XV$. To minimise the computational cost, the eigendecomposition of either the Gram or covariance matrix can be obtained in terms of the decomposition of the other, depending on which of the two matrices is the smallest.[1]

We draw upon this strong connection between the Gram and covariance matrices to consider whether it is possible to avoid the quadratic cost to compute the $N \times N$ attention matrix, which is computed from the analogue of the $N \times N$ Gram matrix $QK^\top{=}XW_qW_k^\top X^\top$. Below we consider how we can use the $d_k \times d_q$ cross-covariance matrix, $K^\top Q{=}W_k^\top X^\top XW_q$, which can be computed in linear time in the number of elements $N$, to define an attention mechanism.

## 3.2   Cross-covariance attention

We propose a cross-covariance based self-attention function that operates along the feature dimension, rather than along the token dimension as in token self-attention. Using the definitions of queries, keys and values from above, the cross-covariance attention function is defined as:

$$\text{XC-Attention}(Q, K, V) = V\,\mathcal{A}_{\mathrm{XC}}(K, Q), \qquad \mathcal{A}_{\mathrm{XC}}(K, Q) = \mathrm{Softmax}\left(\hat{K}^\top \hat{Q}/\tau\right), \qquad (1)$$

where each output token embedding dimension is a convex combination of the $d_v$ features of its corresponding token embedding in $V$. The attention weights $\mathcal{A}$ are computed based on the cross-covariance matrix.

$\ell_2$**-Normalization and temperature scaling.**   In addition to building our attention operation on the cross-covariance matrix, we make a second modification compared to token self-attention. We restrict the magnitude of the query and key matrices by $\ell_2$-normalising them, such that each column of length $N$ of the normalised matrices $\hat{Q}$ and $\hat{K}$ has unit norm, and every element in $d \times d$ cross-covariance matrix $\hat{K}^\top \hat{Q}$ is in the range $[-1, 1]$. We observed that controlling the norm strongly enhances the

---

[1]For $C$ to represent the covariance, $X$ should be centered, *i.e.* $X\mathbf{1}{=}\mathbf{0}$. For the relation between $C$ and $G$, however, centering is not required.

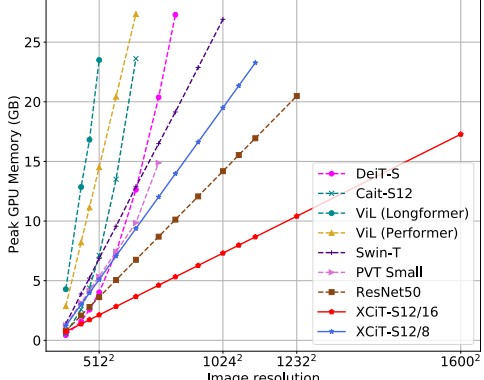
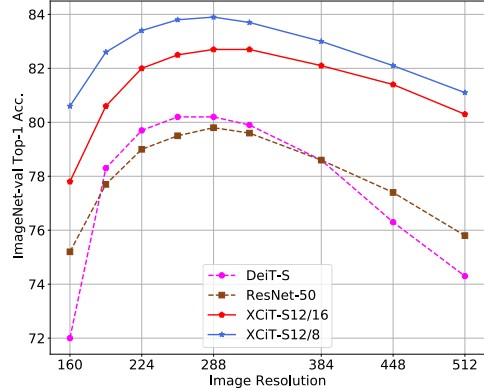

Figure 2: Inference memory usage of vision transformer variants. Our XCiT models scale linearly in the number of tokens, which makes it possible to scale to much larger image sizes, even in comparison to approaches employing approximate self-attention or a pyramidal design. All measurements are performed with a batch size of 64 on a single V100-32GB GPU.

Figure 3: Performance when changing the resolution at test-time for models with a similar number of parameters. All networks were trained at resolution 224, w/o distillation. XCiT is more tolerant to changes of resolution than the Gram-based DeiT and benefit more from the "FixRes" effect [63] when inference is performed at a larger resolution than at train-time.

stability of training, especially when trained with a variable numbers of tokens. However, restricting the norm reduces the representational power of the operation by removing a degree of freedom. Therefore, we introduce a learnable temperature parameter $\tau$ which scales the inner products before the Softmax, allowing for sharper or more uniform distribution of attention weights.

**Block-diagonal cross-covariance attention.** Instead of allowing all features to interact among each other, we divide them into a $h$ groups, or "heads", in a similar fashion as multi-head token self-attention. We apply the cross-covariance attention separately per head where for each head, we learn separate weight matrices to project $X$ to queries, keys and values, and collect the corresponding weight matrices in the tensors $W_q \in \mathbb{R}^{h \times d \times d_q}$, $W_k \in \mathbb{R}^{h \times d \times d_k}$ and $W_v \in \mathbb{R}^{h \times d \times d_v}$, where we set $d_k{=}d_q{=}d_v{=}d/h$. Restricting the attention within heads has two advantages: (i) the complexity of aggregating the values with the attention weights is reduced by a factor $h$; (ii) more importantly, we empirically observe that the block-diagonal version is easier to optimize, and typically leads to improved results. This observation is in line with observations made for Group Normalization [73], which normalizes groups of channels separately based on their statistics, and achieves favorable results for computer vision tasks compared to Layer Normalization [3], which combines all channels in a single group. Figure 4 shows that each head learns to focus on semantically coherent parts of the image, while being flexible to change what type of features it attends to based on the image content.

**Complexity analysis.** The usual token self-attention with $h$ heads has a time complexity of $\mathcal{O}(N^2 d)$ and memory complexity of $\mathcal{O}(hN^2{+}Nd)$. Due to the quadratic complexity, it is problematic to scale token self-attention to images with a large number of tokens. Our cross-covariance attention overcomes this drawback as its computational cost of $\mathcal{O}(Nd^2/h)$ scales linearly with the number of tokens, as does the memory complexity of $\mathcal{O}(d^2/h{+}Nd)$. Therefore, our model scales much better to cases where the number of tokens $N$ is large, and the feature dimension $d$ is relatively small, as is typically the case, in particularly when splitting the features into $h$ heads.

### 3.3 Cross-covariance image transformers

To construct our cross-covariance image transformers (XCiT), we adopt a columnar architecture which maintains the same spatial resolution across layers, similarly to [21, 64, 67]. We combine our cross-covariance attention (XCA) block with the following additional modules, each one being preceded by a LayerNorm [3]. See Figure 1 for an overview. Since in this section we specifically design the model for computer vision tasks, tokens correspond to image patches in this context.

Table 1: **XCiT models**. Design choices include model depth, patch embeddings dimensionality $d$, and the number of heads $h$ used in XCA. By default our models are trained and tested at resolution 224 with patch sizes of 16×16. We also train with distillation using a convolutional teacher (denoted $\Upsilon$) as proposed by Touvron et al. [64]. Finally, we report performance of our strongest models obtained with 8×8 patch size, fine-tuned (↑) and tested at resolution 384×384 (column @384/8), using distillation with a teacher that was also fine-tuned @384.

| Model | Depth | $d$ | #heads | #params | GFLOPs | | ImageNet-1k-val top-1 acc. (%) | | |
|---|---|---|---|---|---|---|---|---|---|
| | | | | | @224/16 | @384/8 | @224/16 | @224/16$\Upsilon$ | @384/8$\Upsilon$ ↑ |
| XCiT-N12 | 12 | 128 | 4 | 3M | 0.5 | 6.4 | 69.9 | 72.2 | 77.8 |
| XCiT-T12 | 12 | 192 | 4 | 7M | 1.2 | 14.3 | 77.1 | 78.6 | 82.4 |
| XCiT-T24 | 24 | 192 | 4 | 12M | 2.3 | 27.3 | 79.4 | 80.4 | 83.7 |
| XCiT-S12 | 12 | 384 | 8 | 26M | 4.8 | 55.6 | 82.0 | 83.3 | 85.1 |
| XCiT-S24 | 24 | 384 | 8 | 48M | 9.1 | 106.0 | 82.6 | 83.9 | 85.6 |
| XCiT-M24 | 24 | 512 | 8 | 84M | 16.2 | 188.0 | 82.7 | 84.3 | 85.8 |
| XCiT-L24 | 24 | 768 | 16 | 189M | 36.1 | 417.9 | 82.9 | 84.9 | 86.0 |

**Local patch interaction.**  In the XCA block communication between patches is only implicit through the shared statistics. To enable explicit communication across patches we add a simple Local Patch Interaction (LPI) block after each XCA block. LPI consists of two depth-wise 3×3 convolutional layers with Batch Normalization and GELU non-linearity in between. Due to its depth-wise structure, the LPI block has a negligible overhead in terms of parameters, as well as a very limited overhead in terms of throughput and memory usage during inference.

**Feed-forward network.**  As is common in transformer models, we add a point-wise feedforward network (FFN), which has a single hidden layer with $4d$ hidden units. While interaction between features is confined within groups in the XCA block, and no feature interaction takes place in the LPI block, the FFN allows for interaction across all features.

**Global aggregation with class attention.**  When training our models for image classification, we utilize the class attention layers as proposed by Touvron et al. [67]. These layers aggregate the patch embeddings of the last XCiT layer through writing to a CLS token by one-way attention between the CLS tokens and the patch embeddings. The class attention is also applied per head, *i.e.* feature group.

**Handling images of varying resolution.**  In contrast to the attention map involved in token self-attention, in our case the covariance blocks are of fixed size independent of the input image resolution. The softmax always operates over the same number of elements, which may explain why our models behave better when dealing with images of varying resolutions (see Figure 3). In XCiT we include additive sinusoidal positional encoding [68] with the input tokens. We generate them in 64 dimensions from the 2d patch coordinates and then linearly project to the transformer working dimension $d$. This choice is orthogonal to the use of learned positional encoding, as in ViT [21]. However, it is more flexible since there is no need to interpolate or fine-tune the network when changing the image size.

**Model configurations.**  In Table 1 we list different variants of our model which we use in our experiments, with different choices for model width and depth. For the patch encoding layer, unless mentioned otherwise, we adopt the alternative used by Graham et al. [26] with convolutional patch projection layers. We also experimented with a linear patch projection as described in [21], see our ablation in Table 4. Our default patch size is 16×16, as in other vision transformer models including ViT [21], DeiT [64] and CaiT [67]. We also experiment with smaller 8×8 patches, which has been observed to improve performance [12]. Note that this is efficient with XCiT as its complexity scales linearly which the number of patches, while ViT, DeiT and CaiT scale quadratically.

# 4   Experimental evaluation

In this section we demonstrate the effectiveness and versatility of XCiT on multiple computer vision benchmarks, and present ablations providing insight on the importance of its different components. In the supplementary material we provide additional analysis, including the impact on performance of image resolution in Section A.1 and of multiple approximate attention baselines in Section A.2.

Table 2: **ImageNet classification**. Number of parameters, FLOPs, image resolution, and top-1 accuracy on ImageNet-1k and ImageNet-V2. Training strategies vary across models, transformer-based models and the reported RegNet mostly follow recipes from DeiT [64].

| Model | #params | FLOPs | Res. | ImNet | V2 |
|---|---|---|---|---|---|
| EfficientNet-B5 RA [17] | 30M | 9.9B | 456 | 83.7 | _ |
| RegNetY-4GF [53] | 21M | 4.0B | 224 | 80.0 | 72.4 |
| DeiT-SϒΥ [64] | 22M | 4.6B | 224 | 81.2 | 68.5 |
| Swin-T [44] | 29M | 4.5B | 224 | 81.3 | _ |
| CaiT-XS24Υ ↑ [67] | 26M | 19.3B | 384 | 84.1 | 74.1 |
| XCiT-S12/16Υ | 26M | 4.8B | 224 | 83.3 | 72.5 |
| XCiT-S12/16Υ ↑ | 26M | 14.3B | 384 | 84.7 | 74.1 |
| XCiT-S12/8Υ ↑ | 26M | 55.6B | 384 | **85.1** | **74.8** |
| EfficientNet-B7 RA [17] | 66M | 37.0B | 600 | 84.7 | _ |
| NFNet-F0 [10] | 72M | 12.4B | 256 | 83.6 | 72.6 |
| RegNetY-8GF [53] | 39M | 8.0B | 224 | 81.7 | 72.4 |
| TNT-B [79] | 66M | 14.1B | 224 | 82.8 | _ |
| Swin-S [44] | 50M | 8.7B | 224 | 83.0 | _ |
| CaiT-S24Υ ↑ [67] | 47M | 32.2B | 384 | 85.1 | 75.4 |
| XCiT-S24/16Υ | 48M | 9.1B | 224 | 83.9 | 73.3 |
| XCiT-S24/16Υ ↑ | 48M | 26.9B | 384 | 85.1 | 74.6 |
| XCiT-S24/8Υ ↑ | 48M | 105.9B | 384 | **85.6** | **75.7** |
| Fix-EfficientNet-B8 [65] | 87M | 89.5B | 800 | 85.7 | 75.9 |
| RegNetY-16GF [53] | 84M | 16.0B | 224 | 82.9 | 72.4 |
| Swin-B↑ [44] | 88M | 47.0B | 384 | 84.2 | _ |
| DeiT-BϒΥ ↑ [64] | 87M | 55.5B | 384 | 85.2 | 75.2 |
| CaiT-S48Υ ↑ [67] | 89M | 63.8B | 384 | 85.3 | **76.2** |
| XCiT-M24/16Υ | 84M | 16.2B | 224 | 84.3 | 73.6 |
| XCiT-M24/16Υ ↑ | 84M | 47.7B | 384 | 85.4 | 75.1 |
| XCiT-M24/8Υ ↑ | 84M | 187.9B | 384 | **85.8** | 76.1 |
| NFNet-F2 [10] | 194M | 62.6B | 352 | 85.1 | 74.3 |
| NFNet-F3 [10] | 255M | 114.8B | 416 | 85.7 | 75.2 |
| CaiT-M24Υ ↑ [67] | 186M | 116.1B | 384 | 85.8 | 76.1 |
| XCiT-L24/16Υ | 189M | 36.1B | 224 | 84.9 | 74.6 |
| XCiT-L24/16Υ ↑ | 189M | 106.0B | 384 | 85.8 | 75.8 |
| XCiT-L24/8Υ ↑ | 189M | 417.8B | 384 | **86.0** | **76.6** |

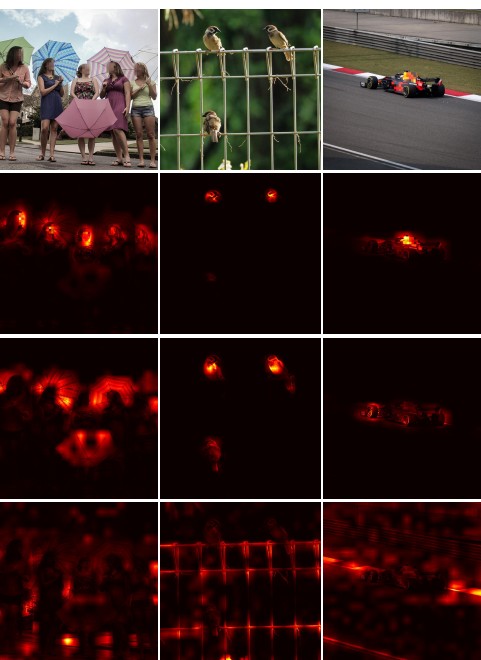

Figure 4: Visualization of the attention map between the CLS token and individual patches in the class-attention stage. For each column, each row represents the attention map w.r.t. one head, corresponding to the image in the first row. Each head appears sensitive to semantically coherent regions. Heads are sensitive to similar features within the same or across images (*e.g.* people or bird faces). They are trigger by different concepts when such features are missing (*e.g.*, cockpit for race cars).

### 4.1 Image classification

We use ImageNet-1k [19] to train and evaluate our models for image classification. It consists of 1.28M training images and 50k validation images, labeled across 1,000 semantic categories. Our training setup follows the DeiT recipe [64]. We train our model for 400 epochs with the AdamW optimizer [45] using a cosine learning rate decay. In order to enhance the training of larger models, we utilize LayerScale [67] and adjust the stochastic depth [33] for each of our models accordingly (see the supplementary material for details). Following [67], images are cropped with crop ratio of 1.0 for evaluation. In addition to the ImageNet-1k validation set, we report results for ImageNet-V2 [55] which has a distinct test set. Our implementation is based on the Timm library [72].

**Results on ImageNet.** We present a family of seven models in Table 1 with different operating points in terms of parameters and FLOPs. We observe that the performance of the XCiT models benefits from increased capacity both in depth and width. Additionally, consistent with [64, 67] we find that using hard distillation with a convolutional teacher improves the performance. Because of its linear complexity in the number of tokens, it is feasible to train XCiT at 384×384 resolution with small 8×8 patches, *i.e.* 2304 tokens, which provides a strong boost in performance across all configurations.

We compare to the state-of-the-art convolutional and transformer-based architectures [10, 44, 53, 58, 67] in Table 2. By varying the input image resolution and/or patch size, our models provide competitive or superior performance across model sizes and FLOP budgets. First, the models operating on 224×224 and 16×16 (*e.g.* XCiT-S12/16) enjoy high accuracy at relatively few FLOPs compared to their counterparts with comparable parameter count and FLOPs. Second, our models with 16×16 and 384×384 resolution images (*e.g.* XCiT-S12/16↑) yield an improved accuracy at the expense of higher FLOPs, and provide superior or on-par performance compared to state-of-the-art

Table 3: **Self-supervised learning.** Top-1 acc. on ImageNet-1k. We report with a crop-ratio 0.875 for consistency with DINO. For the last row it is set to 1.0 (improves from 80.7% to 80.9%). All models are trained for 300 epochs.

| SSL Method | Model | #params | FLOPs | Linear | k-NN |
|---|---|---|---|---|---|
| MoBY [76] | Swin-T [44] | 29M | 4.5B | 75.0 | – |
| DINO [12] | ResNet-50 [28] | 23M | 4.1B | 74.5 | 65.6 |
| DINO [12] | ViT-S/16 [21] | 22M | 4.6B | 76.1 | 72.8 |
| DINO [12] | ViT-S/8 [21] | 22M | 22.4B | **79.2** | **77.2** |
| DINO [12] | XCiT-S12/16 | 26M | 4.9B | 77.8 | 76.0 |
| DINO [12] | XCiT-S12/8 | 26M | 18.9B | **79.2** | 77.1 |
| DINO [12] | ViT-B/16 [21] | 87M | 17.5B | 78.2 | 76.1 |
| DINO [12] | ViT-B/8 [21] | 87M | 78.2B | 80.1 | 77.4 |
| DINO [12] | XCiT-M24/16 | 84M | 16.2B | 78.8 | 76.4 |
| DINO [12] | XCiT-M24/8 | 84M | 64.0B | 80.3 | 77.9 |
| DINO [12] | XCiT-M24/8↑384 | 84M | 188.0B | **80.9** | **78.3** |

Table 4: **Ablations** of various architectural design choices on the task of ImageNet-1k classification using the XCiT-S12 model. Our baseline model uses the convolutional projection adopted from LeVit.

| Model | Ablation | ImNet top-1 acc. |
|---|---|---|
| XCiT-S12/16 | Baseline | 82.0 |
| XCiT-S12/8 | | 83.4 |
| XCiT-S12/16 | Linear patch proj. | 81.1 |
| XCiT-S12/8 | | 83.1 |
| XCiT-S12/16 | w/o LPI layer | 80.8 |
| | w/o XCA layer | 75.9 |
| XCiT-S12/16 | w/o $\ell_2$-normal. | failed |
| | w/o learned temp. $\tau$ | 81.8 |

models with comparable computational requirements. Finally, the linear complexity of XCiT allows us to scale to process $384 \times 384$ images with $8 \times 8$ patch sizes (*e.g.* XCiT-S12/8↑), achieving the highest accuracy across the board, albeit at a relatively high FLOPs count.

**Class attention visualization.** In Figure 4 we show the class attention map obtained in the feature aggregation stage. Each head focuses on different semantically coherent regions in the image (*e.g.* faces or umbrellas). Furthermore, heads tend to focus on similar patterns across images (*e.g.* bird head or human face), but adapts by focusing on other salient regions when such patterns are absent.

**Robustness to resolution changes.** In Figure 3 we report the accuracy of XCiT-S12, DeiT-S and ResNet-50 trained on $224 \times 224$ images and evaluated at different image resolutions. While DeiT outperforms ResNet-50 when train and test resolutions are similar, it suffers from a larger drop in performance as the image resolution deviates farther from the training resolution. XCiT displays a substantially increased accuracy when train and test resolutions are similar, while also being robust to resolution changes, in particular for the model with $8 \times 8$ patches.

**Self-supervised learning.** We train XCiT in a self-supervised manner using DINO [12] on ImageNet-1k. In Table 3 we report performance using the linear and k-NN protocols as in [12]. Across model sizes XCiT obtains excellent accuracy with both protocols, substantially improving DINO with ResNet-50 or ViT architectures, as well as over those reported for Swin-Transformer trained with MoBY [76]. Comparing the larger models to ViT, we also observed improved performance for XCiT achieving a strong 80.3% accuracy. For fair comparison, all reported models have been trained for 300 epochs. Further improved performance of small models is reported by Caron et al. [12] when training for 800 epochs, which we expect to carryover to XCiT based on the results presented here.

**Analysis and ablations.** In Table 4 we provide ablation experiments to analyse the impact of different design choices for our XCiT-S12 model. First, we observe the positive effect of using the convolutional patch projection as compared to using linear patch projection, for both $8 \times 8$ and $16 \times 16$ patches. Second, while removing the LPI layer reduces the accuracy by only 1.2% (from 82.0 to 80.8), removing the XCA layer results in a large drop of 6.1%, underlining the effectiveness of XCA. We noticed that the inclusion of two convolutional components – convolutional patch projection and LPI – not only brings improvements in accuracy, but also accelerates training. Third, although we were able to ensure proper convergence without $\ell_2$-normalization of queries and keys by tweaking the hyper-parameters, we found that it provides stability across model size (depth and width) and other hyper-parameters. Finally, while the learnable softmax temperature parameter is not critical, removing it drops accuracy by 0.2%. Additional ablations are provided in the supplementary material.

Table 5: **COCO object detection and instance segmentation** performance on the mini-val set. All backbones are pre-trained on ImageNet-1k, use Mask R-CNN model [29] and are trained with the same 3x schedule.

| Backbone | #params | $AP^b$ | $AP^b_{50}$ | $AP^b_{75}$ | $AP^m$ | $AP^m_{50}$ | $AP^m_{75}$ |
|---|---|---|---|---|---|---|---|
| ResNet18 [28] | 31.2M | 36.9 | 57.1 | 40.0 | 33.6 | 53.9 | 35.7 |
| PVT-Tiny [70] | 32.9M | 39.8 | 62.2 | 43.0 | 37.4 | 59.3 | 39.9 |
| ViL-Tiny [81] | 26.9M | 41.2 | 64.0 | 44.7 | 37.9 | 59.8 | 40.6 |
| XCiT-T12/16 | 26.1M | 42.7 | 64.3 | 46.4 | 38.5 | 61.2 | 41.1 |
| XCiT-T12/8 | 25.8M | **44.5** | **66.4** | **48.8** | **40.3** | **63.5** | **43.2** |
| ResNet50 [28] | 44.2M | 41.0 | 61.7 | 44.9 | 37.1 | 58.4 | 40.1 |
| PVT-Small [70] | 44.1M | 43.0 | 65.3 | 46.9 | 39.9 | 62.5 | 42.8 |
| ViL-Small [81] | 45.0M | 43.4 | 64.9 | 47.0 | 39.6 | 62.1 | 42.4 |
| Swin-T [44] | 47.8M | 46.0 | 68.1 | 50.3 | 41.6 | 65.1 | 44.9 |
| XCiT-S12/16 | 44.3M | 45.3 | 67.0 | 49.5 | 40.8 | 64.0 | 43.8 |
| XCiT-S12/8 | 43.1M | **47.0** | **68.9** | **51.7** | **42.3** | **66.0** | **45.4** |
| ResNet101 [28] | 63.2M | 42.8 | 63.2 | 47.1 | 38.5 | 60.1 | 41.3 |
| ResNeXt101-32 | 62.8M | 44.0 | 64.4 | 48.0 | 39.2 | 61.4 | 41.9 |
| PVT-Medium [70] | 63.9M | 44.2 | 66.0 | 48.2 | 40.5 | 63.1 | 43.5 |
| ViL-Medium [81] | 60.1M | 44.6 | 66.3 | 48.5 | 40.7 | 63.8 | 43.7 |
| Swin-S [44] | 69.1M | 48.5 | 70.2 | 53.5 | 43.3 | 67.3 | 46.6 |
| XCiT-S24/16 | 65.8M | 46.5 | 68.0 | 50.9 | 41.8 | 65.2 | 45.0 |
| XCiT-S24/8 | 64.5M | 48.1 | 69.5 | 53.0 | 43.0 | 66.5 | 46.1 |
| ResNeXt101-64 [75] | 101.9M | 44.4 | 64.9 | 48.8 | 39.7 | 61.9 | 42.6 |
| PVT-Large [70] | 81.0M | 44.5 | 66.0 | 48.3 | 40.7 | 63.4 | 43.7 |
| ViL-Large [81] | 76.1M | 45.7 | 67.2 | 49.9 | 41.3 | 64.4 | 44.5 |
| XCiT-M24/16 | 101.1M | 46.7 | 68.2 | 51.1 | 42.0 | 65.6 | 44.9 |
| XCiT-M24/8 | 98.9M | **48.5** | **70.3** | **53.4** | **43.7** | **67.5** | **46.9** |

Table 6: **ADE20k semantic segmentation** performance using Semantic FPN [38] and UperNet [74] (in comparable settings). We do not include comparisons with other state-of-the-art models that are pre-trained on larger datasets [44, 54, 83].

| Backbone | Semantic FPN | | UperNet | |
|---|---|---|---|---|
| | #params | mIoU | #params | mIoU |
| ResNet18 [28] | 15.5M | 32.9 | - | - |
| PVT-Tiny [70] | 17.0M | 35.7M | - | - |
| XCiT-T12/16 | 8.4M | 38.1 | 33.7M | 41.5 |
| XCiT-T12/8 | 8.4M | **39.9** | 33.7 | **43.5** |
| ResNet50 [28] | 28.5M | 36.7 | 66.5M | 42.0 |
| PVT-Small [70] | 28.2M | 39.8 | - | - |
| Swin-T [44] | - | - | 59.9M | 44.5 |
| XCiT-S12/16 | 30.4M | 43.9 | 52.4M | 45.9 |
| XCiT-S12/8 | 30.4M | **44.2** | 52.3M | **46.6** |
| ResNet101 [28] | 47.5M | 38.8 | 85.5M | 43.8 |
| ResNeXt101-32 [75] | 47.1M | 39.7 | - | - |
| PVT-Medium [70] | 48.0M | 41.6 | - | - |
| Swin-S [44] | - | - | 81.0M | 47.6 |
| XCiT-S24/16 | 51.8M | 44.6 | 73.8M | 46.9 |
| XCiT-S24/8 | 51.8M | **47.1** | 73.8M | **48.1** |
| ResNeXt101-64 [75] | 86.4M | 40.2 | - | - |
| PVT-Large [70] | 65.1M | 42.1 | - | - |
| Swin-B [44] | - | - | 121.0M | 48.1 |
| XCiT-M24/16 | 90.8M | 45.9 | 109.0M | 47.6 |
| XCiT-M24/8 | 90.8M | **46.9** | 108.9M | **48.4** |

## 4.2 Object detection and instance segmentation

Our XCiT models can efficiently process high-resolution images (see Figure 2). Additionally, XCiT has a better adaptability to varying image resolutions compared to ViT models (see Figure 3). These two properties make XCiT a good fit for dense prediction tasks including detection and segmentation.

We evaluate XCiT for object detection and instance segmentation using the COCO benchmark [42] which consists of 118k training and 5k validation images including bounding boxes and mask labels for 80 categories. We integrate XCiT as backbone in the Mask R-CNN [29] detector with FPN [43]. Since the XCiT architecture is inherently columnar, we make it FPN-compatible by extracting features from different layers, *e.g.*, layers 4, 6, 8, and 12 for XCiT-S12. All features have a constant stride of 8 or 16 based on the patch size, and the feature resolutions are adjusted to have strides of 4, 8, 16, and 32, similar to ResNet-FPN backbones, where the downsampling is achieved by max pooling and the upsampling is obtained using a single transposed convolution layer (see the supplementary material for details). The model is trained for 36 epochs (3x schedule) using the AdamW optimizer with learning rate of $10^{-4}$, 0.05 weight decay and 16 batch size. We adopt the multiscale training and augmentation strategy of DETR [11]. Our implementation is based on the mmdetection library [13].

**Results on COCO.** In Table 5 we report object detection and instance segmentation results of four variants of XCiT using $16 \times 16$ and $8 \times 8$ patches. We compare to ResNets [28] and concurrent efficient vision transformers [44, 70, 81]. All models are trained using the 3x schedule after ImageNet-1k pre-training. Note that other results with higher absolute numbers have been achieved when pre-training on larger datasets [44] or with longer schedules [4], and are therefore not directly comparable to the reported results. First, across all model sizes XCiT outperforms the convolutional ResNet [28] and ResNeXt [75] by a large margin with either patch size. Second, we observe a similar increase in accuracy compared to PVT [70] and ViL [81] backbones. Finally, XCiT provides a competitive performance with Swin [44].[2] For relatively small models, XCiT-S12/8 outperforms its Swin-T counterpart with a decent margin. On the other hand, Swin-S provides slightly stronger results compared to XCiT-S24/8. Utilizing smaller $8 \times 8$ patches leads to a consistent gain across all models.

---

[2]We use report the results provided by the authors in their open-sourced code https://github.com/SwinTransformer/Swin-Transformer-Object-Detection.

### 4.3 Semantic segmentation

We further show transferability of our models with semantic segmentation experiments on the ADE20k dataset [84], which consists of 20k training and 5k validation images with labels over 150 semantic categories. We integrate our backbones in two segmentation methods: Semantic FPN [38] and UperNet [74]. We train for 80k and 160k iterations for Semantic FPN and UperNet respectively. Following [44], the models are trained using batch size 16 and an AdamW optimizer with learning rate of $6 \times 10^{-5}$ and 0.01 weight decay. We apply the same method of extracting FPN features as explained in Section 4.2. We report the performance using the standard single scale protocol (without multi-scale and flipping). Our implementation is based on the mmsegmentation library [16].

**Results on ADE20k.** We present the semantic segmentation performance using XCiT backbones in Table 6. First, for Semantic FPN [38], XCiT provides a superior performance compared to ResNet, ResNeXt and PVT backbones using either option of patch size. Second, compared to Swin Transformers using the same UperNet decoder [74], XCiT with 8×8 patches consistently achieves a higher mIoU for different models. XCiT with 16×16 patches provides a strong performance especially for smaller models where XCiT-S12/16 outperforms Swin-T.

## 5 Conclusion

**Contributions.** We present an alternative to token self-attention which operates on the feature dimension, eliminating the need for expensive computation of quadratic attention maps. We build our XCiT models with the cross-covariance attention as its core component and demonstrate the effectiveness and generality of our models on various computer vision tasks. In particular, it exhibits a strong image classification performance on par with state-of-the-art transformer models while similarly robust to changing image resolutions as convnets. XCiT is effective as a backbone for dense prediction tasks, providing excellent performance on object detection, instance and semantic segmentation. Finally, we showed that XCiT can be a strong backbone for self-supervised learning, matching the state-of-the-art results with less compute. XCiT is a generic architecture that can readily be deployed in other research domains where self-attention has shown success.

**Limitations.** Our models enable training with smaller patches and on higher-resolution images, which leads to clear performance gains. However, for tasks like image classification this gain comes at a cost of relatively high number of FLOPs. In order to address this issue, other components, like FFN, could also be re-examined. Another point is that XCiT models seem to overfit more than their CaiT counterparts, see Table 2. They are more similar to some convnets in that respect.

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
