# XCiT: Cross-Covariance Image Transformers

## Appendix

## A    Preliminary study on Vision Transformers (ViT)

In this appendix we report the results associated with our preliminary study on high-resolution transformers. Most of the experiments were carried out on the ViT architecture [21] with DeiT training [64], and intended to analyze different aspects of transformers when considering images with varying resolution or high-resolution images specifically.

### A.1    Impact of resolution versus patch size

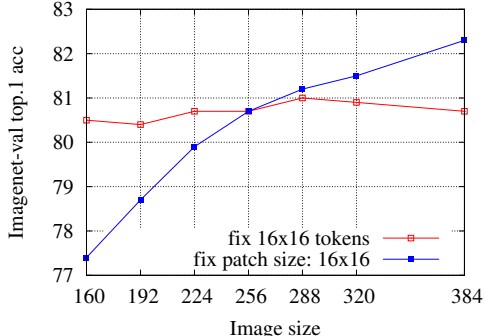

| Variable patch size | | | | | | |
|---|---|---|---|---|---|---|
| Image Size | 80 | 112 | 160 | 256 | 320 | 384 |
| Patch Size | 5 | 7 | 10 | 16 | 20 | 24 |
| Top-1 | 78.2 | 79.7 | 80.5 | 80.7 | 80.9 | 80.7 |

| Variable number of tokens | | | | | | |
|---|---|---|---|---|---|---|
| Image Size | 160 | 224 | 256 | 288 | 320 | 384 |
| # of tokens | 100 | 196 | 256 | 324 | 400 | 576 |
| Top-1 | 77.4 | 79.9 | 80.7 | 81.2 | 81.5 | 82.3 |

Figure A.1: **Impact of input resolution on accuracy for DeiT-S**. We consider different image resolutions, and either (1) increase the patch size while keeping the number of tokens fixed; or (2) keep the patch size fixed and use more tokens. Larger input images are beneficial if the number of tokens increases. The impact of a change of a resolution for a constant number of patches (of varying size) is almost neutral. As one can observe, the main driver of performance is the number of patches. The patch size has a limited impact on the accuracy, except when considering very small ones. We have observed and confirmed similar trends with XCiT models.

### A.2    Approximate attention models in ViT with DeiT training

In Table A.1, we report the results that we obtain by replacing the Multi-headed Self-attention operation with efficient variants [30, 56, 69, 70] in the DeiT-S backbone. First, we can notice that for all efficient self-attention choices there is a clear drop in performance compared to the Deit-S baseline. The spatial reduction attention (SRA) proposed in PVT [70] has a significantly weaker performance compared to the full-attention with a quadratic complexity that is more efficient than full-attention by only a constant factor $R^2$. Linformer [69] provides a better accuracy compared to SRA, however, it is also clearly weaker than full-attention. Moreover, Linformer does not have the flexibility of processing variable length sequences which limits its application in many computer vision tasks. Efficient attention [56] provides a better trade-off than the aforementioned methods, with improved accuracy and linear complexity. However, it has a 3.6% drop in performance compared to full-attention. Finally, axial attention [30] provides the strongest performance among the efficient attention variants we studied with a 1.5% drop in accuracy compared to the baseline. We observe a saving in memory usage, but a drop in speed due to the separate row and column attention operations. Our observations are consistent with [21].

### A.3    Training and testing with varying resolution

As discussed in the main manuscript, for several tasks it is important that the network is able to handle images of varying resolutions. This is the case, for instance, for image segmentation, object detection, or image retrieval where the objects of interest may have very different sizes. We present an analysis of train/test resolution trade-off in Table A.2.

I

Table A.1: **ImageNet Top-1 accuracy of efficient self-attention variants** (after 300 epochs of training).

| Model | Complexity | Top-1 |
|---|---|---|
| DeiT-S [64] | $\mathcal{O}(N^2)$ | 79.9 |
| SRA (Average Pool) [70] | $\mathcal{O}(N^2/R^2)$ | 73.5 |
| SRA (Convolutional) [70] | $\mathcal{O}(N^2/R^2)$ | 74.0 |
| Linformer (k=$\sqrt{n}$) [69] | $\mathcal{O}(kN)$ | 75.7 |
| Efficient Transformer [56] | $\mathcal{O}(N)$ | 76.3 |
| Axial [30] | $\mathcal{O}(N\sqrt{N})$ | 78.4 |

Table A.2: **Trade-off between train and test resolutions for DeiT.** MS refers to multi-scale training, where the models have seen images from different resolutions at training time.

| Test / Train | 160 | 224 | 256 | 288 | 320 | MS |
|---|---|---|---|---|---|---|
| 160 | **77.2** | 75.9 | 73.3 | 68.2 | 59.6 | 76.3 |
| 224 | 78.0 | **79.9** | 79.9 | 79.0 | 77.9 | 79.6 |
| 256 | 77.3 | 80.4 | **80.7** | 80.2 | 79.9 | 80.6 |
| 288 | 76.3 | 80.4 | 81.0 | **81.2** | 80.8 | 81.0 |
| 320 | 75.0 | 80.1 | 80.9 | 81.3 | **81.5** | 81.3 |

# B Additional details of training and our architecture

## B.1 Sinusoidal positional encoding

We adopt a sinusoidal positional encoding as proposed by Vaswani et al. [68] and adapted to the 2D case by Carion et al. [11]. However we depart from this method in that we first produce this encoding in an intermediate 64d space before projecting it to the working space of the transformers. More precisely, in our implementation each of the $x$ and $y$ coordinates is encoded using 32 dimensions corresponding to cosine and sine functions, with 16 different frequencies for each function. The encoding of both coordinates are eventually concatenated to obtain a 64 dimension 2d positional encoding. Finally, the 64 dimension positional encoding is linearly projected to the working dimension of the model $d$.

## B.2 Obtaining feature pyramid for dense prediction

For state-of-the-art detection and segmentation models, FPN is an important component which provides features of multiple scales. We adapt XCiT to be compatible with FPN detection and segmentation methods through a simple re-scaling of the features extracted from different layers. In particular, for models with 12 layers, we extract features from the 4[th], 6[th], 8[th] and 12[th] layer respectively. As for models with 24 layers, we extract features from the 8[th], 12[th], 16[th] and 24[th] layer. Concerning the re-scaling of the features, the 4 feature levels are downsized by a ratio of 4, 8, 16 and 32 compared to the input image size. Feature downsizing is performed with max pooling and upsampling is achieved using a single layer of transposed convolutions with kernel size $k = 2$ and stride $s = 2$.

## B.3 Hyper-parameters: LayerScale initialization and Stochastic Depth drop-rate

We list the stochastic depth $d_r$ and LayerScale initialization $\epsilon$ hyperparameters used by each of our models in Table B.1.

# C Pseudo-code

In Algorithm 1 we provide a PyTorch-style pseudo code of the Cross-covariance attention operation. The pseudo code resembles the Timm library [72] implementation of token self-attention. We show that XCA only requires few modifications, namely the $\ell_2$ normalization, setting the learnable temperature parameters and a transpose operation of the keys, queries and values.

Table B.1: **Hyperparameters used for training our models**, including the Stochastic depth drop rate $d_r$ and LayerScale initialization $\epsilon$.

| Model | Patch size | $d_r$ | $\epsilon$ |
|---|---|---|---|
| XCiT-N12 | 8 & 16 | 0.0 | 1.0 |
| XCiT-T12 | 8 & 16 | 0.0 | 1.0 |
| XCiT-T24 | 8 & 16 | 0.05 | $10^{-5}$ |
| XCiT-S12 | 8 & 16 | 0.05 | 1.0 |
| XCiT-S24 | 8 & 16 | 0.1 | $10^{-5}$ |
| XCiT-M24 | 8 & 16 | 0.15 | $10^{-5}$ |
| XCiT-L24 | 16 | 0.25 | $10^{-5}$ |
| XCiT-L24 | 8 | 0.3 | $10^{-5}$ |

---

**Algorithm 1** Pseudocode of XCA in a PyTorch-like style.

```
# self.qkv: nn.Linear(dim, dim * 3, bias=qkv_bias)
# self.temp: nn.Parameter(torch.ones(num_headss, 1, 1))

def forward(self, x):
    B, N, C = x.shape
    qkv = self.qkv(x).reshape(B, N, 3, self.num_heads, C // self.num_heads)
    qkv = qkv.permute(2, 0, 3, 1, 4)
    q, k, v = qkv[0], qkv[1], qkv[2] # split into query, key and value

    q = q.transpose(-2, -1)
    k = k.transpose(-2, -1) # Transpose to shape (B, h, C, N)
    v = v.transpose(-2, -1)

    q = F.normalize(q, dim=-1, p=2) # L2 Normalization across the token dimension
    k = F.normalize(k, dim=-1, p=2)

    attn = (k @ q.transpose(-2, -1)) # Computing the block diagonal cross-covariance matrix
    attn = attn * self.temp # Adjusting the activations scale with temperature parameter
    attn = attn.softmax(dim=-1) # d x d attention map

    x = attn @ v # Apply attention to mix channels per token
    x = x.permute(0, 3, 1, 2).reshape(B, N, C)
    x = self.proj(x)
    return x
```

---

# D    Additional results

## D.1    More XCiT models

We present additional results for our XCiT models in Table D.1. We include performance of $384{\times}384$ images using a $16{\times}16$ patch size as well as results for images with $224{\times}224$ resolution using patch size of $8{\times}8$.

Table D.1: **ImageNet-1k top-1 accuracy of XCiT** for additional combinations of image and patch sizes.

| Models | Depth | $d$ | #Blocks | params | $16 \times 16$ patches | | | | $8 \times 8$ patches | | | |
|---|---|---|---|---|---|---|---|---|---|---|---|---|
| | | | | | GFLOPs | @224 | @224$\Upsilon$ | @384$\uparrow$ | GFLOPs | @224 | @224$\Upsilon$ | @384$\uparrow$ |
| XCiT-N12 | 12 | 128 | 4 | 3M | 0.5 | 69.9 | 72.2 | 75.4 | 2.1 | 73.8 | 76.3 | 77.8 |
| XCiT-T12 | 12 | 192 | 4 | 7M | 1.2 | 77.1 | 78.6 | 80.9 | 4.8 | 79.7 | 81.2 | 82.4 |
| XCiT-T24 | 24 | 192 | 4 | 12M | 2.3 | 79.4 | 80.4 | 82.6 | 9.2 | 81.9 | 82.6 | 83.7 |
| XCiT-S12 | 12 | 384 | 8 | 26M | 4.8 | 82.0 | 83.3 | 84.7 | 18.9 | 83.4 | 84.2 | 85.1 |
| XCiT-S-24 | 24 | 384 | 8 | 48M | 9.1 | 82.6 | 83.9 | 85.1 | 36.0 | 83.9 | 84.9 | 85.6 |
| XCiT-M24 | 24 | 512 | 8 | 84M | 16.2 | 82.7 | 84.3 | 85.4 | 63.9 | 83.7 | 85.1 | 85.8 |
| XCiT-L24 | 24 | 768 | 16 | 189M | 36.1 | 82.9 | 84.9 | 85.8 | 142.2 | 84.4 | 85.4 | 86.0 |

## D.2    Transfer learning

In order to further demonstrate the flexibility and generality of our models, we report transfer learning experiments in Table D.2 for models that have been pre-trained using ImageNet-1k and finetuned for other datasets including CIFAR-10, CIFAR-100 [40], Flowers-102 [47], Stanford Cars [39] and iNaturalist [31]. We observe that the XCiT models provide competitive performance when compared to strong baselines like ViT-B, ViT-L, DeiT-B and EfficientNet-B7.

Table D.2: **Evaluation on transfer learning.**

| Architecture | CIFAR$_{10}$ | CIFAR$_{100}$ | Flowers102 | Cars | iNat$_{18}$ | iNat$_{19}$ |
|---|---|---|---|---|---|---|
| EfficientNet-B7 [58] | 98.9 | **91.7** | **98.8** | 94.7 | _ | _ |
| ViT-B/16 [21] | 98.1 | 87.1 | 89.5 | _ | _ | _ |
| ViT-L/16 [21] | 97.9 | 86.4 | 89.7 | _ | _ | _ |
| Deit-B/16 [64] ϒ | **99.1** | 91.3 | **98.8** | 92.9 | 73.7 | 78.4 |
| XCiT-S24/16 ϒ | **99.1** | 91.2 | 97.4 | 92.8 | 68.8 | 76.1 |
| XCiT-M24/16 ϒ | **99.1** | 91.4 | 98.2 | 93.4 | 72.6 | 78.1 |
| XCiT-L24/16 ϒ | **99.1** | 91.3 | 98.3 | 93.7 | **75.6** | **79.3** |

## D.3 Image retrieval

**Context of this study.** Vision-based retrieval tasks such as landmark or particular object retrieval have been dominated in the last years by methods extracting features from high-resolution images. Traditionally, the image description was obtained as the aggregation of local descriptors, like in VLAD [36]. Most of the modern methods now rely on convolutional neural networks [6, 24, 60]. In a recent paper, El-Nouby *et al.* [22] show promising results with vision transformers, however they also underline the inherent scalability limitation associated with the fact that ViT models do not scale well with image resolution. Therefore, it cannot compete with convolutional neural networks whose performance readily improve with higher resolution images. Our XCiT models do not suffer from this limitation: our models scale linearly with the number of pixels, like convnets, and therefore makes it possible to use off-the-shelf methods initially developed for retrieval with high-resolution images.

### D.3.1 Datasets and evaluation measure

In each benchmark, a set of query images is searched in a database of images and the performance is measured as the mean average precision.

The Holidays [35] dataset contains images of 500 different objects or scenes. We use the version of the dataset where the orientation of images (portrait or landscape) has been corrected. Oxford [49] is a dataset of building images, which corresponds to famous landmark in Oxford.

Table D.3: **The basic statistics on the image retrieval datasets.**

| Dataset | number of images | | nb of instances |
|---|---|---|---|
| | database | queries | |
| Holidays | 1491 | 500 | 500 |
| R-Oxford | 4993 | 70 | 26 |

We use the revisited version of the Oxford benchmark [51], which breaks down the evaluation into easy, medium and hard categories. We report results on the "medium" and "hard" settings, as we observed that the ordering of techniques does not change under the easy measures.

### D.3.2 Image representation: global and local description with XCiT

We consider three existing methods to extract an image vector representations from the pre-trained XCiT models. Note that to the best of our knowledge, for the first time we extract local features from the output layer of a transformer layer, and treat them as patches fed to traditional state-of-the-art methods based on matching local descriptors or CNN.

**CLS token.** Similar to El-Nouby et al. [22] with ViT, we use the final vector as the image descriptor. In this context, the introduction of class-attention layers can be regarded as a way to learn the aggregation method.

Table D.4: **Instance retrieval experiments.** The default resolution is 768. The default class token size is 128 dimensions. The "local descriptor" representation extracted from the activations is in 128 dimensions. To our knowledge the state of the art with ResNet-50 on Holidays with Imagenet pre-training only is the Multigrain method [6], which achieves mAP=92.5%. Here we compare against this method under the same training setting, i.e., off-the-shelf network pre-trained on ImageNet-1k only and with the same training procedure and resolution. We refer the reader to Tolias *et al.* [61] for the state of the art on R-Oxford, which involves some training on the target domain with images depicting building and fine-tuning at the target resolution.

| Base model | parameters | R-Oxford5k (mAP) | | Holidays (mAP) |
|---|---|---|---|---|
| | | Medium | Hard | |
| **XCiT– class token** | | | | |
| XCiT-S12/16 | | 30.1 | 8.7 | 86.0 |
| XCiT-S12/8 | | 33.2 | 12.1 | 86.4 |
| XCiT-S12/16 | resolution 224 | 12.7 | 2.4 | 71.5 |
| XCiT-S12/16 | resolution 384 | 20.1 | 4.6 | 83.4 |
| XCiT-S12/16 | resolution 512 | 26.6 | 5.8 | 84.6 |
| XCiT-S12/16 | resolution 768 | 30.1 | 8.7 | 86.0 |
| XCiT-S12/16 | resolution 1024 | 30.3 | 11.2 | 86.3 |
| XCiT-S12/16 | self-supervised DINO | 35.1 | 11.9 | 87.3 |
| XCiT-S12/8 | self-supervised DINO | 30.9 | 7.9 | 88.3 |
| **XCiT– VLAD** | | | | |
| XCiT-S12/16 | k=256 | 36.6 | 11.6 | 89.9 |
| XCiT-S12/16 | k=1024 | 40.0 | 13.0 | 90.7 |
| **XCiT– ASMK** | | | | |
| XCiT-S12/8 | k=1024 | 36.5 | 9.4 | 90.4 |
| XCiT-S12/8 | k=65536 | 42.0 | 12.9 | 92.3 |
| XCiT-S12/16 | k=1024 | 35.2 | 11.5 | 90.4 |
| XCiT-S12/16 | k=65536 | 40.0 | 15.0 | 92.0 |
| **ResNet-50 – ASMK** | | | | |
| ResNet50 | k=1024 | 41.6 | 14.6 | 86.0 |
| ResNet50 | k=65536 | 41.9 | 14.5 | 87.9 |
| Multigrain-resNet50 | k=1024 | 32.9 | 9.4 | 87.9 |

**VLAD.**     We treat the patches before the class-attention layers as individual local descriptors, and aggregate them into a higher-dimensional vector by employing the Vector of locally aggregated Descriptors [36].

**ASMK.**     We also apply the aggregated selective match kernel from Tolias *et al.* [59]. This method was originally introduced for local descriptors, but got adapted to convolutional networks. To the best of our knowledge this is the state of the art on several benchmarks [61].

For all these methods, we use the models presented in our main paper, starting from the version fine-tuned at resolution 384×384. By default the resolution is 768. This is comparable to the choice adopted in the literature for ResNet (e.g., 800 in the work by Berman et al. [6]).

### D.3.3   Experimental setting: Image retrieval with models pretrained on ImageNet-1k only

We only consider models pre-trained on Imagenet-1k. Note that the literature reports significant improvement when learning or fine-tuning networks [52, 61] on specialized datasets (e.g., of buildings for Oxford5k). We consider only XCiT-S12 models, since they have a number of parameters comparable to that of ResNet-50. We report the results in Table D.4.

**Scaling resolution.**     As expected increasing the resolution with XCiT improves the performance steadily up to resolution 768. This shows that our models are very tolerant to resolution changes considering that they have been fine-tuned at resolution 384. The performance starts to saturates at resolution 1024, which led us to keep 784 as the pivot resolution.

**Self-supervision.** The networks XCiT pre-trained with self-supervision achieve a comparatively better performance than their supervised counterpart on Holidays, however, we have the opposite observation for R-Oxford.

**Impact of image description.** We adopt the class-token as the descriptor, and in our experiments we verified that this aggregation method is better than average and GeM pooling [8, 52]. In Table D.4 one can see there is a large benefit in employing a patch based method along with our XCiT transformers: XCiT-VLAD performs significantly better than the CLS token, likely thanks to the higher dimensionality. This is further magnified with AMSK, where we obtain results approaching the absolute state of the art on Holidays, despite a sub-optimal training setting for image retrieval. This is interesting since our method has not been fine-tuned for retrieval tasks, and has not been adapted in any significant way beyond applying off-the-shelf aggregation techniques. A direct comparison with ResNet-50 shows that our XCiT method obtains competitive results in this comparable setting, slightly below the ResNet-50 on R-Oxford but significantly better on Holidays.

### D.4 Runtime and memory usage

We present the peak memory usage as well as the throughput of multiple models including full-attention and efficient vision transformers in Table D.5. Additionally, in Figure D.1 we plot the processing speed represented as millisecond per image as a function of image resolution for various models. We can observe that XCiT provides a strong trade-off, possessing the best scalability in terms of peak memory, even when compared to ResNet-50. Additionally, the processing time scales linearly with respect to resolution, with only ResNet-50 providing a better trade-off on that front.

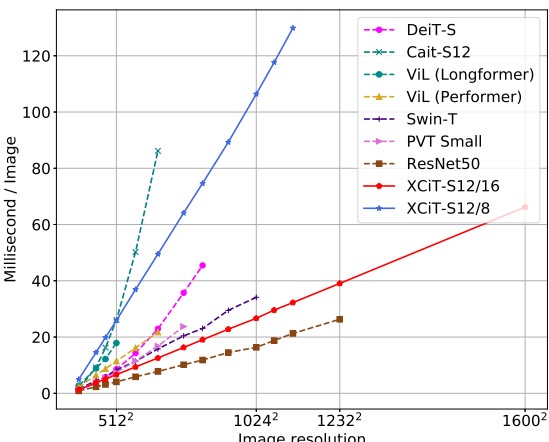

Figure D.1: **Throughput in millisecond per image during inference of multiple models.** Our XCiT-S12/16 model provides a speed up for images with higher resolution compared to existing vision transformers, especially the ones with quadratic complexity like DeiT and CaiT.

Table D.5: **Inference throughput and peak GPU memory usage** for our XCiT small model compared to other models of comparable size that include token self-attention. All models tested using batch size of 64 on a V100 GPU with 32GB memory.

| Model | #params ($\times 10^6$) | ImNet Top-1 @224 | Image Resolution | | | | | | | |
|---|---|---|---|---|---|---|---|---|---|---|
| | | | $224^2$ | | $384^2$ | | $512^2$ | | $1024^2$ | |
| | | | im/sec | mem (MB) | im/sec | mem (MB) | im/sec | mem (MB) | im/sec | mem (MB) |
| ResNet-50 | 25 | 79.0 | 1171 | 772 | 434 | 2078 | 245 | 3618 | 61 | 14178 |
| DeiT-S | 22 | 79.9 | 974 | 433 | 263 | 1580 | 116 | 4020 | N/A | OOM |
| CaiT-S12 | 26 | 80.8 | 671 | 577 | 108 | 2581 | 38 | 7117 | N/A | OOM |
| PVT-Small | 25 | 79.8 | 777 | 1266 | 256 | 3142 | 134 | 5354 | N/A | OOM |
| Swin-T | 29 | 81.3 | 704 | 1386 | 220 | 3890 | 120 | 6873 | 29 | 26915 |
| XCiT-S12/16 | 26 | 82.0 | 781 | 731 | 266 | 1372 | 151 | 2128 | 37 | 7312 |

VI

## D.5 Query and key magnitude visualizations

$$\|\hat{Q}\| \qquad \|\hat{K}\|$$

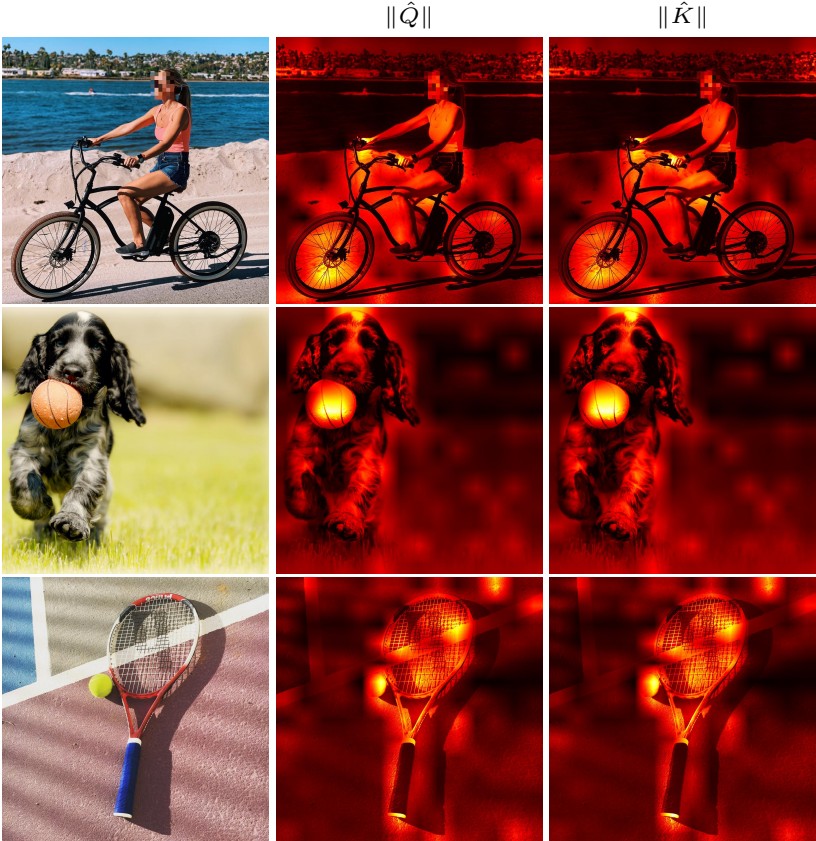

Figure D.2: **Visualization of the queries $\hat{Q}$ and keys $\hat{K}$ norm across the feature dimension.** We empirically observe that magnitude of patch embeddings in the queries and keys correlates with the saliency of their corresponding region in the image.

Our XCA operation relies on the cross-covariance matrix of the queries $\hat{Q}$ and keys $\hat{K}$ which are $\ell_2$ normalized across the patch dimension. Therefore, each element in the $d \times d$ matrix represents a cosine similarity whose value is strongly influenced by the magnitude of each patch. In Figure D.2 we visualize the magnitude of patch embeddings in the queries and keys matrices. We observe that patch embeddings with higher magnitude corresponds to more salient regions in the image, providing interpretable visualization of which regions in the image contribute more in the cross-covariance attention.