# OpenReview forum: "XCiT: Cross-Covariance Image Transformers"
_NeurIPS.cc/2021/Conference — NeurIPS 2021 Poster_

### Official Review · Reviewer_DFVp · 2021-07-18

**Rating:** 7
**Confidence:** 3

**Summary:**

This paper proposes an efficient method to compute self-attention in Transformers without sacrificing accuracy. The idea is to compute the covariance self-attention over the feature dimension followed by a 3x3 conv to model the patch interaction. With additional techniques such as feature grouping and L2 norm, the method shows competitive accuracy on ImageNet compared to the recent Vision Transformer models. This method has linear complexity to the input patches and thus can be used for high-resolution images on the detection and segmentation tasks.

**Limitations And Societal Impact:**

Yes

**Main Review:**

**Originality**: High the method is new in vision transformers.

**Quality**: Good. The method is well motivated and explained. The main claim appears to be supported by strong empirical results.

**Clarity**: Great.

**Significance**: High. This fundamental method enables linear computation of self-attention over high-resolution images which may impact many works in object detection and segmentation.

Strengths:
+ Novel and efficient method that is well-motivated by covariance computation and 1x1 conv operator.
+ The problem is non-trivial for which the authors develop additional techniques in order to obtain promising results.
+ Extensive experiments are conducted over multiple tasks to support its efficiency and accuracy.

Weaknesses:
- The performance on ImageNet is not state-of-the-art without distillation. It is okay but is it fair to include the distillation numbers in Table 2 as many of the baselines do not? What is the performance for the teacher conv model for the models in Table 1?

- The comparison to other efficient attention computation methods (in Table A of Appendix) should be included in the main paper. Would the results still be consistent on other popular Vision Transformers other than DeiT?

- As the advantage of the method is on high-resolution images, why not compare with other baselines in Table A on object detection or segmentation?

My preliminary rating is 7 but I hope the authors can address the above questions.


=====
Post-rebuttal

Thank you for answering the questions. My questions are mainly about clarity and presentation, and the author made them clearer in the rebuttal. I keep my rating and hope the authors would make the changes stated in the rebuttal.

**Time Spent Reviewing:**

3.5

---

> ### Author Response · Authors · 2021-08-10
> **Response to Reviewer DFVp**
>
> We thank the reviewer for their feedback and questions. We address the weaknesses mentioned by the reviewer in the following points.
>
> **Q1: The performance on ImageNet is not state-of-the-art without distillation. It is okay but is it fair to include the distillation numbers in Table 2 as many of the baselines do not? What is the performance for the teacher conv model for the models in Table 1?**
>
> We report results without distillation in Table 1. To simplify the comparison, we will add non-distilled results to Table 2.
>
>
>
> **Q2: The comparison to other efficient attention computation methods (in Table A of Appendix) should be included in the main paper. Would the results still be consistent on other popular Vision Transformers other than DeiT?**
>
>
> We will try to include results in Tables A.1-A.2 into the main paper despite the space limitations. Regarding other vision transformers, we already study performance of  PVT, Swin and ViL in the main paper, which represent the most competitive approximate attention transformers in computer vision tasks. We will be happy to consider comparisons with other transformer models if suggested by the reviewer.
>
> **Q3: As the advantage of the method is on high-resolution images, why not compare with other baselines in Table A on object detection or segmentation?**
>
> We thank the reviewer for the suggestion. We will consider adding these results to the final version of the paper.

---

### Official Review · Reviewer_d8h9 · 2021-07-19

**Rating:** 6
**Confidence:** 5

**Summary:**

This paper presents a modification of vision transformers by introducing Cross-Covariance Attention (XCA) and Local Patch Interaction (LPI) which substantially reduce the computational overhead and enhances the stability of training. The authors also adopt the DeiT training strategy with distillation using a convolutional teacher to improve the performance. Experiments on Imagenet classification, self-supervised training with DINO and many downstream tasks (COCO object detection and instance segmentation, ADE20K semantic segmentation) show the superior performance.

**Limitations And Societal Impact:**

See above

**Main Review:**

The core idea of Cross-Covariance Attention (XCA) to make transformer linearity in this paper was originally proposed in [56]. This submission use it to make the whole transformer model work with L2-Normalization and temperature scaling.
On the other hand, the Local Patch Interaction (LPI) is good but do not bring any theoratical contribution or give exciting intuition despite the 1.2% improvement on ImageNet (Table 4).
Another contribution is that the final model performs better than original ViT on various tasks including self-supervised training with DINO and many downstream tasks.
It also achieves state-of-the-art performance by using the DeiT training strategy.
However, this paper does not bring any new idea but incorporates several different things together. I vote for rejection.

[56] Zhuoran Shen, Mingyuan Zhang, Haiyu Zhao, Shuai Yi, and Hongsheng Li. Efficient attention: Attention with linear complexities. In Proceedings of the IEEE/CVF Winter Conference on Applications of Computer Vision, 2021.

**Time Spent Reviewing:**

4

---

> ### Author Response · Authors · 2021-08-10
> **Response to Reviewer d8h9**
>
> We thank the reviewer for their feedback. Below we clarify the difference of our work with respect to [56] and further emphasize our contributions.
>
> **Q1: The core idea of Cross-Covariance Attention (XCA) to make transformer linearity in this paper was originally proposed in [56]. This submission use it to make the whole transformer model work with L2-Normalization and temperature scaling.**
>
> Similar to other recent works, Efficient Attention (EA) in [56] aims to avoid explicit computation of the NxN attention matrix (see L.98-99). Both EA [56] and our XCA have linear complexity in N. The motivation and the formulation of XCA, however, is substantially different with respect to EA. We explain these differences below and provide experimental evidence demonstrating the practical advantage of XCA over [56].
>
> Conceptually, EA attempts to approximate the self-attention operation in the original Transformer. In contrast, our XCA operation is motivated by the relationship between the cross-covariance and the Gram matrix. We specifically study the attention over the cross-covariance matrix of different channels. We, hence, emphasize that XCA is not an approximation of the Gram-based self-attention, which is the case for [56].
>
> In terms of formulations, EA performs a softmax row-wise normalization for the values and a column-wise softmax for the keys. In contrast, XCA applies a softmax normalization to the inner product of Keys and Queries. Although the difference of the two formulations may appear subtle, it leads to substantially different computations followed by the difference in experimental results. Below we clarify the relation of EA and XCA formulations in more detail.
>
> XCA is defined as:
>
> (1) $\text{XCA} = V \text{Softmax}(||K||^T ||Q|| / \tau).$
>
> Efficient attention (the softmax version used as default in the paper) is defined as:
>
> (2) $\text{EA} = \text{Softmax}_r(Q)( \text{Softmax}_c (K)^T V)$
>
> We note that the dxd matrix is obtained using the queries Q and keys K for XCA, but using Keys K and Values V for EA. Assuming the assignment of names is arbitrary, we re-define Efficient attention by swapping Queries and Values:
>
> (3) $\text{EA} = \text{Softmax}_r(V)( \text{Softmax}_c (K)^T Q)$
>
> By comparing (1) and (3) we can now better see the difference between XCA and EA. Clearly, the application of a non-linear softmax operation to V and K in the case of EA differs substantially from the softmax applied to the product of K^T and Q in the case of XCA
>
> Experimentally, we demonstrate the superior performance of XCiT when compared to strong baselines on multiple tasks, including classification, detection and segmentation (see Tables 2-6). To compare our approach with EA [56], we conduct an additional experiment where we replace XCA by the Efficient Attention (w/ softmax). For fair comparison we use the same experimental settings and report results of XCiT using either XCA or EA blocks. As we can see from the table below, XCA outperforms EA by 1.1% in top-1 accuracy which is a significant gain for the ImageNet benchmark. This clearly confirms the advantage of our proposed XCA formulation.
>
> | Model | IN-1k Top-1 Accuracy |
> |---------:|:-----------|
> | XCiT w/ XCA | 82.0% |
> | XCiT w/ EA (softmax) | 80.9%|
>
> We further emphasize the difference of our method to [56] by the following.
> + [56] has been proposed as an augmentation to ConvNets, similar to Non-local blocks. In contrast, we show that XCA can be used as the core building block for a vision transformer. We show that even without the LPI component, only XCA can provide an 80.8% performance on ImageNet (see Table 4).
> + In [56], the authors observe low position sensitivity of attention maps: “Each k_{j} is a global attention map that does not correspond to any specific position. Instead, each of them corresponds to a semantic aspect of the entire input.” In contrast, our Keys and Queries are both sensitive to image locations as they highlight salient regions in the image (see Figure D.2). This further indicates that XCA and EA perform different operations.
>
> **Q2: On the other hand, the Local Patch Interaction (LPI) is good but do not bring any theoratical contribution or give exciting intuition despite the 1.2% improvement on ImageNet (Table 4).**
>
> We motivate LPI in Section 3.3 of the paper. LPI is a simple and efficient method that enables explicit cross-patch communication and compensates for the lack of such communication in the XCA module. Our ablation in Table 4  shows that LPI adds 1.2% improvement in performance, which is a significant gain for ImageNet top-1 accuracy.
>
>
>
> **Q3: Another contribution is that the final model performs better than original ViT on various tasks including self-supervised training with DINO and many downstream tasks. It also achieves state-of-the-art performance by using the DeiT training strategy. However, this paper does not bring any new idea but incorporates several different things together.**
>
>
> We respectfully disagree regarding the lack of novelty in our paper. Our proposed XCA block is substantially different from previous methods and can be used to build stand-alone transformer models. While being more efficient, such models retain and improve the performance of explicit patch-to-patch attention models, including recent transformer models specialized for particular tasks. XCiT provides a new and efficient solution to scale vision transformers to high resolution images. Moreover, we demonstrate that XCiT is more robust to changes in image resolution compared to its ViT/DeiT counterparts. Our paper and supplementary material provide comprehensive analysis, ablations and visualizations supporting our claims and explaining the intuition behind our method.

---

> > ### Comment · Reviewer_d8h9 · 2021-08-30
> > **followup**
> >
> > Thanks for the response. It's clear that the proposed XCA is different to EA [56]. I will upgrade the rating.

---

### Official Review · Reviewer_oXoV · 2021-07-25

**Rating:** 7
**Confidence:** 5

**Summary:**

This paper proposes the idea of bypassing the calculation of NxN attention scores and directly deriving the weighted average of values in a self-attention by first calculating DxD cross-covariance matrix of K^TQ and then averaging over the D features of V instead of its tokens. Since this method of attention does not involve cross-token communication, the paper further proposes to use a Local Patch Interaction module in which a depth-wise 3x3 convolution enables cross-token communication.

**Limitations And Societal Impact:**

The societal impact has not been discussed in the paper. However, the limitations are mentioned and discussed.

**Main Review:**

Manuscript: The paper is very well-written and easy to follow. While the paper provides results on multiple tasks and datasets, the comparison to state-of-the-art is not complete and very little ablation is provided.

Novelty: The core idea of the paper is interesting. However, intuitive justification for some ideas is missing in the paper. It is important to cover the following questions in the paper:
1. The intuition behind a conventional attention layer is clear. What is the equivalent intuition behind XCA attention layer?
2. The paper claims that adding LPI helps with accuracy and convergence. This might be due to the Batch Normalization in this layer. Have the authors tried only a Batch Normalization (which also implicitly brings cross-sample stats in) after the XCA modules?
3. The paper argues against MLP-Mixer for deriving attention scores by learnable weights. Related to this argument, an interesting ablation to consider is applying a channel-wise MLP instead of LPI after the attention module to observe the effect of such projection. This, while still quadratic in computation, is necessary to support such argument. It essentially shows what happens if we replace the MLP-2 module in MLP-Mixer with an XCA.

**Time Spent Reviewing:**

5

---

> ### Author Response · Authors · 2021-08-10
> **Response to Reviewer oXoV**
>
> We would like to thank the reviewer for their feedback and suggestions. We address the reviewer’s questions in the following and we will update the paper with more clarifications concerning these points in the final version.
>
> **Q1: The intuition behind a conventional attention layer is clear. What is the equivalent intuition behind XCA attention layer?**
>
>
> The XCA layer exploits the cross-covariance of two projections of the patch representation (i.e. Keys and Queries) at each layer through an attention operation and dynamically generates 1D filters. Such filters are then used to update the representation of each patch. This can be thought of as an advanced, attention-based squeeze-and-excitation operation.
>
>
> **Q2: The paper claims that adding LPI helps with accuracy and convergence. This might be due to the Batch Normalization in this layer. Have the authors tried only a Batch Normalization (which also implicitly brings cross-sample stats in) after the XCA modules?**
>
> As per the reviewer's suggestion, we experimented with replacing the LPI module by the BatchNorm layer. Below we report performance for three different settings: (1) When using no LPI component; (2) When using BatchNorm and (3) When using the complete LPI component. One can see that the BatchNom adds 0.3% to the accuracy while LPI improves the no-LPI version by 1.2%. This suggests that the BatchNorm layer is indeed useful, however, the LPI module provides additional gain due to convolutional layers.
>
> | Model | IN-1k Top-1 Accuracy |
> |---------:|:----------------------------|
> | XCiT w/o LPI | 80.8% |
> | XCiT w/ BN | 81.1% |
> | XCiT w/ LPI | 82.0% |
>
> **Q3: The paper argues against MLP-Mixer for deriving attention scores by learnable weights. Related to this argument, an interesting ablation to consider is applying a channel-wise MLP instead of LPI after the attention module to observe the effect of such projection. This, while still quadratic in computation, is necessary to support such argument. It essentially shows what happens if we replace the MLP-2 module in MLP-Mixer with an XCA.**
>
> Following the reviewer’s suggestion, we conduct an experiment where we replace the LPI module by an MLP-1 block from the MLP-Mixer. As we can see from the table below, the resulting model with 27M parameters achieves 81.5% top-1 ImageNet accuracy. The most comparable MLP-Mixer model (in term of params),  Mixer-B/16 with 59M parameters achieves significantly lower accuracy of 76.4% despite using more parameters. This experiment clearly shows the advantage of XCA compared to MLP-Mixer. By comparing the two XCiT results below we can also observe that the LPI component provides a better option for cross-patch communication compared to MLP-1. We will add this ablation to the final version of the paper
>
>
> | Model | #Params | IN-1k Top-1 Accuracy |
> |---------:|:-----------:|:-----------------|
> | XCiT w/ LPI | 26M | 82.0% |
> | XCiT w/ MLP-1 | 27M | 81.5% |
> | MLP-Mixer-B/16 | 59M | 76.4% |
>
> We also note that the MLP models, similar to Transformers, are typically learned for a fixed image size. In contrast, our approach shows excellent behavior across different image resolutions, as we report in Figure 3. This property is useful for many vision tasks involving images of various sizes and aspect ratios, which is typically the case for object detection and semantic segmentation.
>
>
> We will add these additional experiments and ablations to the paper, and hope they answer the reviewers questions. We will be happy to answer any further questions.

---

### Official Review · Reviewer_nWMF · 2021-07-26

**Rating:** 7
**Confidence:** 5

**Summary:**

This paper proposes a “transposed” self-attention module, it has linear complexity in the token side, which is more efficient on high-resolution images. The performance of XCiT on ImageNet classification  is promising, it also shows strong results on detection/segmentation and self-supervised pre-training.

**Limitations And Societal Impact:**

Yes

**Main Review:**

This paper has clear paper writing, nice reference/related work, and very solid experiments. The novelty is also good for me.

Here are some suggestions:

This paper seems like a new version of SENet[1], the proposed Cross-Covariance Attention (XCA) actually does a self-attention at channel level, which is more complex than SENet's way (and the results is better).  It achieves spatial interaction with 3x3 dw-convs. It makes sense.

The idea of XCA is good. It can reduce the complexity of self-attention. But I have several questions. If the authors can solve my concerns in rebuttal, I am happy to increase my score.

(1) How about the results on detection/segmentation of combining XCA with general-purpose Vision Transformers such as PVT[2] and Swin Transformer[3]?
Because XCiT can only return features with the same scales, and manually resize them to multiple scales, which is not elegant and may hurt performance.
This paper has a strong potential to handle high-resolution images, which is important for object detection and semantic segmentation tasks. Choosing PVT or Swin as a baseline is a better choice.

(2) How about using regular self-attention for patch interaction instead of using 3x3 convs? Can it further improve the performance? Or use efficient attention in PVT(sparse manner) or Swin(local manner).

(3) Miss the ablation study about comparison XCA and Squeeze-and-Excitation.


Reference

[1] Squeeze-and-Excitation Networks

[2] Pyramid Vision Transformer: A Versatile Backbone for Dense Prediction without Convolutions

[3] Swin Transformer: Hierarchical Vision Transformer using Shifted Windows

**Time Spent Reviewing:**

3

---

> ### Author Response · Authors · 2021-08-10
> **Response to Reviewer nWMF**
>
> We thank the reviewer for their constructive feedback. We address the raised points and suggestions below.
>
> **Q1: How about the results on detection/segmentation of combining XCA with general-purpose Vision Transformers such as PVT[2] and Swin Transformer[3]? Because XCiT can only return features with the same scales, and manually resize them to multiple scales, which is not elegant and may hurt performance. This paper has a strong potential to handle high-resolution images, which is important for object detection and semantic segmentation tasks. Choosing PVT or Swin as a baseline is a better choice.**
>
> The PVT and Swin Transformers rely on an architecture design inspired by ResNets, where the majority of the model parameters and computations are concentrated in the later layers with low spatial resolution (e.g. $N$=7x7=49) and high patch-embedding dimensionality $d$  (e.g. 1024 or 2048). XCiT, on the other hand, has a linear complexity in the number of tokens N, but quadratic complexity in the dimensionality $d$. Adopting the PVT/Swin design, hence, will reduce benefits provided by XCiT since the majority of computations $O(d^2)$ will be dedicated to layers with d >> N (e.g. 1024 >> 49).
>
> As we report in the paper (Tables 2, 5 and 6), while keeping the simple ``columnar'' non-pyramidal design of the original ViT and NLP Transformers, XCiT achieves results competitive with SoTA and outperforms PVT and Swin by a large margin on the image classification task. Moreover, without a special architecture design dedicated to the dense prediction tasks, XCiT also outperforms PVT and Swin in semantic segmentation and object detection for the majority of operating points (Tables 5,6). We believe that XCiT provides a good balance between the accuracy and efficiency and demonstrates that the original design of columnar Transformers can be used for dense high-resolution prediction tasks.
>
> **Q2: How about using regular self-attention for patch interaction instead of using 3x3 convs? Can it further improve the performance? Or use efficient attention in PVT(sparse manner) or Swin(local manner).**
>
> The goal of LPI is to provide a very efficient method, both in parameters and compute, to allow explicit communication between patches. A full self-attention could achieve that, however, it would diminish efficiency gains provided by the XCA and would break the linear complexity in $N$ of XCiT.
>
> An approximate version of self-attention as suggested by the reviewer is an interesting alternative. We have followed this suggestion and performed an experiment replacing the LPI module by a SRA (R=8) from PVT. From results below we observe that SRA provides a weaker performance compared to LPI while increasing the model’s parameter count significantly.
>
> | Model   |      #Params     |     IN-1k  Top-1  Accuracy  |
> |----------------:|:----------:|:---------|
> |XCiT w/ LPI  | 26M  | 82.0% |
> | XCiT w/ SRA (R=8)| 146M | 81.1% |
>
>
> **Q3: Miss the ablation study about comparison XCA and Squeeze-and-Excitation.**
>
> We thank the reviewer for an interesting suggestion.We have tried replacing the XCA component with a Squeeze and Excitation module. While the SE version of XCiT provides a relatively decent performance, XCA improves the performance by 1% on top-1 accuracy of ImageNet, which is a significant improvement for this metric. We will include this ablation in the final version of the paper.
>
> | Model   |   IN-1k  Top-1  Accuracy  |
> |----------------:|:----------|
> |XCiT w/ XCA   | 82.0% |
> | XCiT w/ SE | 81.0% |
>
> We hope our clarifications answer the reviewer’s questions. We will be happy to address any further questions and to include additional insights to the paper.

---

### Decision · Program_Chairs · 2021-09-27

**Decision:**

Accept (Poster)

**Comment:**

Initially, the paper received three acceptance and one rejection. One reviewer had concern on the novelty of the paper given its similarity with EA. Upon rebuttal and discussion, the reviewer converged with the other reviewers on the novelty of this paper. The AC agrees with the reviewers and recommends to accept the paper. The authors are encouraged to improve final version of the paper following the suggestions from the reviewers.